# Usage Patterns of Traditional Chinese Medicine for Patients with Bipolar Disorder: A Population-Based Study in Taiwan

**DOI:** 10.3390/healthcare12040490

**Published:** 2024-02-18

**Authors:** Shu-Ping Chen, Su-Tso Yang, Kai-Chieh Hu, Senthil Kumaran Satyanarayanan, Kuan-Pin Su

**Affiliations:** 1Graduate Institute of Chinese Medicine, School of Chinese Medicine, College of Chinese Medicine, China Medical University, Taichung 404328, Taiwan; u108058202@cmu.edu.tw; 2School of Chinese Medicine, College of Chinese Medicine, China Medical University, Taichung 404328, Taiwan; 3Management Office for Health Data, China Medical University Hospital, Taichung 404439, Taiwan; 4College of Medicine, China Medical University, Taichung 404328, Taiwan; cobol@cmu.edu.tw; 5Mind-Body Interface Laboratory (MBI-Lab), China Medical University Hospital, Taichung 404327, Taiwan

**Keywords:** bipolar disorder, herbal medicine, traditional Chinese medicine, Taiwan’s National Health Insurance Research Database

## Abstract

Background: Patients with bipolar disorder (BD) receive traditional Chinese medicine (TCM) for clinical needs unmet with psychotropic medications. However, the clinical characteristics of practices and outcomes of TCM in BD are not fully understood. This cohort study investigated the clinical characteristics, principal diagnoses, TCM interventions, and TCM prescriptions in patients with BD. Methods: Data for a total of 12,113 patients with BD between 1996 and 2013 were withdrawn from Taiwan’s longitudinal health insurance database 2000 (LHID 2000). The chi-square test was used for categorical variables, and the independent *t*-test was used for continuous variables. A *p*-value less than 0.05 indicated significance. Results: One thousand three hundred nineteen patients who visited TCM clinics after the diagnosis of BD were in the TCM group, while those who never visited TCM were in the non-TCM group (*n* = 1053). Compared to the non-TCM group, patients in the TCM group had younger average age, a higher percentage of female individuals, more comorbidities of anxiety and alcohol use disorders, and higher mood stabilizer usage rates. The TCM group exhibited pain-related indications, including joint pain, myalgia, myositis, headache, and sleep disturbances. Corydalis yanhusuo and Shu-Jing-Huo-Xue-Tang were the most useful single herbs and herbal formulae. Conclusions: Physicians need to be aware of the use of TCM in patients with BD.

## 1. Introduction

Bipolar disorder is a mental disorder that includes three episodes: mania, hypomania, and depression [1]. The prevalence of bipolar disorder (BD) is about two to five percent of the worldwide population [2]. Over 90% of patients with BD experience recurrence throughout their lives, and approximately 35 to 57% relapse in one year [3]. The treatment goal for BD is to control acute episodes that, when severe, may lead to hospitalization and to prevent future recurrence states [4,5]. Mood stabilizers (e.g., lithium, valproate acid, carbamazepine, and antipsychotics) are approved to suppress shifts between mania and depression. Monotherapy of mood stabilizers should be the primary choice, at least in mild and moderate mania. Although poly-therapy with combinations of mood stabilizers plus atypical antipsychotics has proven to be more effective, poly-therapy should be reserved for severe mania or as a subsequent step in mild and moderate mania after unsuccessful monotherapy [6]. Some patients with BD need in-hospital care besides conventional pharmacotherapy. However, pharmacotherapy with lithium only improves the BD prognosis in one-third of patients [7]. When their medications did not relieve manic or depressive symptoms, patients with BD tended to use complementary and alternative medicine (CAM) [8]. Another reason patients with BD used CAM was that it cost less than psychiatric services [9]. Worthy of attention, most patients with BD did not discuss their CAM usage with their physicians [10]. Two papers identified the population of patients with BD using CAM. Fifty percent of 435 patients with BD took vitamins or herbs, and 44% of 50 elderly patients with BD used CAM [10,11].

There is more research on CAM treatments for depression or bipolar depression than there is for CAM treatments for bipolar disorder. According to a review study, CAM therapies for BD include omega-3 fatty acids, St. John’s wort, herbal products, S-adenosyl-L-methionine, aromatherapy massage, therapeutic massage, yoga, Chinese oral medicine, and acupuncture [12]. St. John’s wort can relieve mild to moderate depression but has the potential to induce mania because it can induce cytochrome P450 enzyme activity, accelerating the metabolism of antipsychotics and reducing antipsychotics’ control of mania and hypomania [13]. S-adenosyl-L-methionine can treat major depressive disorder but induces mania with an unclear mechanism [14,15]. Evidence regarding the benefits of omega-3 fatty acids or acupuncture is inconsistent [12].

Although CAM is a broader term encompassing diverse healing practices from various cultural and medical traditions worldwide, the National Center for Complementary and Integrative Health (NCCIH) classified CAM treatments by their primary therapeutic input, how the therapy is taken in or delivered, as follows: nutritional (e.g., special diets, dietary supplements, herbs, and probiotics), psychological (e.g., mindfulness), physical (e.g., massage, spinal manipulation), and combined therapies, such as mental and physical therapy (e.g., yoga, tai chi, acupuncture, dance or art therapies) or psychological and nutritional therapy (e.g., mindful eating) [16]. Traditional Chinese medicine (TCM) is an essential part of CAM. TCM has its roots in ancient Chinese philosophy, including principles such as Yin and Yang and the concept of Qi (vital energy). TCM interventions include acupuncture, herbal medicine, Chinese traumatology medicine, and qigong.

TCM has been used for thousands of years in China. *Huang Di Nei Jing* is an ancient Chinese medical text treated as a fundamental doctrinal source for Chinese medicine. Chapter 46 of *Huang Di Nei Jing Su Wan* and Chapter 22 of *Ling Su* describe symptoms and treatments associated with BD, such as fresh iron flakes and acupuncture on the spleen, bladder, stomach, lung, small intestine, and large intestine meridians [17,18]. Recent research about TCM on BD conducted before April 2022 is as follows: A clinical trial demonstrated that patients with BD taking Free and Easy Wanderer Plus and carbamazepine had lower drop-out rates, side effects, and serum carbamazepine levels than the carbamazepine plus placebo group [19]. A case report also inferred that acupuncture combined with mood stabilizers and antipsychotics as maintenance therapies would not aggravate bipolar disorder prognosis [20].

Research about the population of TCM users and treatments of BD is insufficient. The probable reason is that the number of TCM users among patients with BD in published studies is ambiguous, which cannot support subsequent TCM clinical studies on BD. Therefore, investigating the number of people with BD using TCM is a priority research effort. In this study, we aimed to investigate usage patterns of TCM for patients with BD in Taiwan, including analyzing the characteristics of patients with BD who used TCM, the primary indication of patients with BD visiting TCM clinicians, and the TCM techniques and prescriptions they accept.

## 2. Materials and Methods

### 2.1. Data Source

This study used data from the Longitudinal Health Insurance Database 2000 (LHID2000), which contains the claim data of one million randomly selected patients in the National Health Insurance Research Database (NHIRD) of the single-payer healthcare system launched in Taiwan from 1996 to 2013 (releasing two hundred thousand and eight hundred thousand insured in 2002 and 2009), but not those born after 2000 [21]. The LHID data include patients’ demographics, inpatient and outpatient visit records, prescriptions, and disease diagnoses, encoded based on the International Classification of Diseases, Ninth Revision, Clinical Modification (ICD-9-CM). All patient data had been de-identified to preserve patient privacy. The structure of the NHIRD files is described in detail on the NHIRD website and in other publications [22,23]. The National Health Insurance Research Database (NHIRD) presents all TCM outpatient activities but not inpatient services. NHIRD also covers expenses related to TCM interventions, encompassing oral Chinese herbal medicines (single herbs and herbal formulae), acupuncture, and Chinese traumatology medicine [24]. In the current study, we extracted all TCM users with BD but analyzed Chinese herbal medicine and acupuncture prescriptions in subsequent analysis. This study was approved by the Research Ethics Committee of China Medical University and Hospital [CMUH104-REC2-115(AR-4)], Taichung, Taiwan, 12 June 2020.

### 2.2. Study Population

The inclusion criteria were as follows: both male and female patients with BD (ICD-9-CM: 296.0, 296.1, 296.4, 296.5, 296.6, 296.7, and 296.8) older than fifteen years old in the LHID 2000. Age at onset (AAO) is a critical variable in the prognoses of BD [25]. There are several cut-off BD onset ages in the literature. For example, forty years old is a cut-off for early-onset and late-onset bipolar disorder (EOBD and LOBD) [26]. Approximately 30–60% of BD patients experience onset between ages 15 and 19, called pediatric or juvenile BD [27,28]. The condition of suffering or maintaining BD at over 50 years of age was named older age bipolar disorder (OABD) [29]. Bolton et al., in 2021, systematically searched the databases. They proposed the trimodal age-at-onset distribution with an average early-onset age of seventeen, mid-onset age of twenty-six, and late-onset age of forty-two [25]. Depending on the previous literature statements and the fact that citizens over 65 years old in Taiwan are elderly, we decided to include BD patients more than 15 years old. In total, 12,113 patients diagnosed with BD from 1 January 1996 to 31 December 2013 were included in this study. The index date was the initial visit to a TCM clinic. The current study followed each patient in the cohort until death, withdrawal from the National Health Insurance program, or 31 December 2013. Of these patients, eighty-nine percent (*n* = 10,758) received TCM. Among TCM users with BD, eighty-seven percent (*n* = 9361) visited TCM clinics before and 1397 after BD diagnosis. To avoid potential TCM interference before the diagnosis of BD, we were more interested in exploring the phenomenon of patients who never received TCM treatment before the diagnosis of BD visiting TCM physicians for help not quitting psychiatric therapies. Therefore, all eligible participants in this study were excluded if they were recorded visiting TCM clinics before their diagnosis of BD. Other exclusion criteria were index date not between the enrollment and withdrawal date, index year not between 1996 and 2012, under the age of fifteen, and missing end date data. After the diagnosis of BD, 1319 of the included patients received TCM during conventional medicine (CM) treatment, and 1053 patients only received CM treatments. Figure 1 presents a visualization of the sample selection procedure.

### 2.3. Covariates of Comorbidities and Medications

Physical diseases are risk factors for psychiatric disorders [30,31,32]; psychiatric disorders also affect physical illnesses. When patients’ physical conditions and psychiatric disorders are not controlled well, both will increase the disease prognoses bilaterally [33]. Cognitive functions, pharmacotherapies, and psychiatric and medical comorbidities of patients with BD affect recurrence rates, increasing the odds of suicidality, disability, unemployment, and re-hospitalization [34]. According to previous statements, we selected the following comorbidities in BD patients based on a literature review: Physical comorbidities in patients with BD included diabetes mellitus (ICD-9-CM: 250) [35], hypertension (ICD-9-CM: 401–405) [36], hyperlipidemia (ICD-9-CM: 272) [37], chronic obstructive pulmonary disease (ICD-9-CM: 490–496) [32,38,39], tobacco use (ICD-9-CM: 305.1) [40], and obesity (ICD-9-CM: 278) [41]. The previous comorbid diseases were risk factors for cardiovascular disease, such as ischemic heart disease (ICD-9-CM: 410–414) [42], and one of the most common premature mortalities among BD patients [43,44].

Patients with BD may also have one or several psychiatric diseases as comorbidities. A meta-analysis study indicated that 40% of patients with BD suffer from an anxiety disorder [45,46]. Seventeen percent of BD patients had attention-deficit/hyperactivity disorder (ADHD) [47]. McElroy et al. surveyed 875 patients about their eating disorder comorbidities, and 14.3% had at least one comorbid lifetime eating disorder, such as bulimia nervosa [48]. Regier et al. found 43.6% of patients with BD had an alcohol diagnosis [49]. Therefore, we also included anxiety (ICD-9-CM: 300.0, 300.2, and 300.3), attention-deficit disorder (ICD-9-CM: 314.0), unspecified eating disorder (ICD-9-CM: 307.50), bulimia nervosa (ICD-9-CM: 307.51), and alcohol use disorder (291, 303, 305.00, 305.01, 305.02, and 305.03) as psychiatric comorbidities in patients with BD.

Conventional psychotropic medications such as mood stabilizers and antipsychotics are typically used to manage BD. We selected BD medicines mentioned in Psych Notes- Clinical Pocket Guide 4th Edition by Darlene D. Pedersen and “Taiwan Consensus of Pharmacological Treatment for Bipolar Disorder” by Bai, Y. M. et al. (2013), such as lithium carbonate, carbamazepine, valproic acid, lamotrigine, topiramate, aripiprazole, and loxapine, as treatment-as-usual medications for further analysis [6,50]. The medicines mentioned previously are paid for in BD treatment by Taiwan’s National Health Insurance and are suitable for conducting NHIRD research. All patients with BD received “treatment-as-usual medications” during the study period. The current study was retrospective research, so patients’ medical records do not include interference from outside the NHI during the study period.

### 2.4. Statistical Analysis

Descriptive statistics for categorical and continuous variables are presented as numbers or percentages and means and standard deviations, respectively. Chi-square and Student’s *t*-tests were adopted to evaluate differences in categorical and continuous variables, respectively, between the cohorts. A *p*-value less than 0.05 indicated significance in any hypothesis test. SAS 9.4 (SAS Institute Inc., Cary, NC, USA) was used for data analysis.

## 3. Results

### 3.1. Demographic Characteristics

Table 1 summarizes all participants’ demographic and clinical characteristics (TCM and non-TCM groups). The mean age of patients in the TCM group was less than that of the non-TCM group (44.3 ± 16.8 vs. 48.2 ± 19.0; *p* < 0.0001). The non-TCM group included more men (63.3%) than women, whereas the TCM group included more women (53.4%) than men. The follow-up duration was 8.6 ± 4.8 years in the non-TCM group and 8.3 ± 4.3 years in the TCM group. Significant differences in the comorbidities of anxiety (*p* < 0.0001) and alcoholism (*p* = 0.0128) were noted between the TCM group (31.9% and 5.5% with anxiety and alcoholism, respectively) and the non-TCM group (20.7% and 3.3% with anxiety and alcoholism, respectively). Moreover, a significantly higher proportion of patients in the TCM group took medication, including lithium carbonate, carbamazepine, valproic acid, and aripiprazole.

### 3.2. TCM Treatment Interventions

From 1996 to 2013, 10,758 (88.8%) patients with BD visited TCM clinics. Among these patients, 9361 received TCM treatment before the diagnosis of BD, and 1319 received TCM treatment after the diagnosis of BD. In the latter group of 1319 patients, 1043 (79.1%) took single herbs, 1155 (87.6%) took herbal formulae, and 120 (9.1%) received acupuncture treatment (Table 1).

### 3.3. Principal Diagnostic Codes Used by TCM Practitioners

Table 2 lists the top ten indications for Chinese herbal medicines and acupuncture among patients with BD. The indications identified by TCM doctors are arranged by diagnosis frequency in descending order. For patients who took Chinese herbal medication, indications were sleep disturbances, joint pain, acute nasopharyngitis, unspecified myalgia and myositis, lumbago, cough, headache, contusion of the ankle and foot (excluding toes), lumbar sprain, and constipation. For patients who received acupuncture, indications were joint pain, unspecified myalgia and myositis, lumbago, contusion of the knee and lower leg, ankle sprain, sprains and strains at unspecified sites of the shoulder and upper arm, contusion of the elbow and forearm, contusion of the wrist and hand(s) (except fingers), sprains and strains at unspecified sites of the knee and leg, and unspecified backache.

### 3.4. Top Ten Most Commonly Prescribed Chinese Single Herbs and Herbal Formulae

Table 3 presents the top ten Chinese herbal medicines prescribed to patients with BD. The prescribed single herbs and herbal formulae are arranged by diagnosis frequency in descending order. Single herbs prescribed were yanhusuo (Corydalis yanhusuo W. T. Wang), danshen (Salvia miltiorrhiza Bunge), huangqin (Scutellaria baicalensis Georgi), gegen (Pueraria lobate), jiegeng (Platycodon grandiflorum), gancao (Glycyrrhiza uralensis Fisch.), suanzaoren (Ziziphus jujube Mill. var. spinosa), duzhong (Eucommis ulmoidea Oliv.), dahuang (Rheum palmatum L.), and baizhi (Angelica dahurica). Herbal formulae prescribed were Shu-Jing-Huo-Xue-Tang, Jia-Wei-Xiao-Yao-San, Shao-Yao-Gan-Cao-Tang, Suan-Zao-Ren-Tang, Ge-Gen-Tang, Xue-Fu-Zhu-Yu-Tang, Tian-Wang-Bu-Xin-Dan, Chai-Hu-Jia-Long-Gu-Mu-Li-Tang, Gan-Mai-Da-Zao-Tang, and Chuan-Xiong-Cha-Tiao-San.

Table 4 shows the top 10 Chinese herbal medicines prescribed to patients with sleep disturbances after the diagnosis of BD in Taiwan during 1996–2012. Yejiaoteng (Polygonum multiflorumThunb) and Suan-Zao-Ren-Tang, also contained in Table 3, are the most commonly prescribed prescriptions for sleep disturbance in patients with BD. Other prescriptions such as danshen (Salvia miltiorrhiza Bunge), gegen (Pueraria lobate), suanzaoren (Ziziphus jujube Mill. var. spinosa), Jia-Wei-Xiao-Yao-San, Tian-Wang-Bu-Xin-Dan, Chai-Hu-Jia-Long-Gu-Mu-Li-Tang, and Gan-Mai-Da-Zao-Tang are included in both Table 3 and Table 4.

## 4. Discussion

### 4.1. Demographic Characteristics

Compared with the non-TCM group, demographic characteristics of the TCM group were younger age, female, higher percentages of anxiety and alcohol use disorder, and higher medication usage rates. We highly suspect that patients with more severe illnesses of BD tended to receive TCM treatment. Patients with early-onset BD (age ≤ 18 years) exhibit more severe conditions, a shorter mood cycle, a higher likelihood of having comorbid anxiety, and more substance use [51]. However, we could not determine whether the onset of BD was earlier among the patients in the TCM group than among those in the non-TCM group because NHIRD does not contain data on the age of BD onset. Screening for new diagnoses of BD in the NHIRD was possible, but the age at the time of diagnosis may differ from the onset age.

A systematic review calculated that the lifetime and current prevalence rate of any anxiety disorder comorbidities was 40.5% and 38.2% among patients with BD [45]. In the present study, we figured out that 27% of patients with BD had anxiety states (ICD-9-CM: 300.0), phobic disorders (ICD-9-CM: 300.2), or obsessive–compulsive disorders (ICD-9-CM: 300.3), and more than two-thirds of them received TCM treatment. We found out that patients with BD visiting TCM clinicians may relate to anxiety. Lithium has an essential role in treating comorbid anxiety in BD in patients who were medication-free primarily and lithium-naïve at baseline [52]. However, patients with high anxiety scores had low lithium responsivity [53]. In the current study, we could not determine whether the severity of anxiety disorders from the NHIRD data has a positive correlation with visiting TCM. BD and alcohol use disorder are often contemporary [54]. A sizeable epidemiological study revealed that 43.6% of patients with BD had alcohol use disorder [49]. In our study, only 4.5% of patients with BD had alcohol misuse, and more than 67% visited TCM clinics. However, because of the small sample size of this study, this percentage is not representative of all patients with BD in the real world of Taiwan. Anxiety was a predictor of a shorter time to recurrence, and alcohol abuse was a predictor of a long time to remission [55]. Anxiety and alcohol misuse may be confounding factors in comparing BD severity between patients who did or did not receive TCM treatment.

Psychiatric physicians prescribe two or more medications based on the BD patients’ disease severity when the first-line monotherapy medicine fails to affect them [6]. Our study indicates that significantly more patients with BD in the TCM group than in the non-TCM group took valproate, lithium, and carbamazepine to manage their illnesses. In particular, aripiprazole was prescribed only in the TCM group. However, we did not calculate the impact of combination/augmentation or adjunctive/add-on therapy, which needs future scrutiny.

### 4.2. Acupuncture Treatment Intervention

TCM has included acupuncture treatment for over three thousand years [56]. Acupuncture amplified the American people’s interest in the early 1970s [57]. A study analyzing 22,512 adults in the United States 2007 National Health Interview Survey revealed that 6.8% reported lifetime use of acupuncture and 1.5% reported use in the past 12 months [58]. Taiwan’s NHIRD from 1996 to 2002 showed that 22.6% of total valid beneficiaries (*n* = 4,948,464) used acupuncture during the study period [59]. Another comparative study analyzed the 2002 to 2011 data of Taiwan’s LHID 2000 and found that the one-year population of acupuncture users increased from 7.98% to 10.9% [60]. Further calculations indicated that the average acupuncture population was 8.84% during the study period [60]. Our analysis also used LHID 2000 and found that 9% of patients with BD received acupuncture treatment between 1996 and 2012, which seemed higher than the average, but there is a need for more comprehensive statistics to support our findings. In our study, patients with BD visiting TCM clinics tended to take herbal medicines rather than receive acupuncture. There are two possible reasons. First, patients with BD probably cannot overcome the fear of needles as acupuncture is a slightly painful invasive treatment that causes soreness in the treatment area [61]. A clinical study indicated that the fear of pain induced by acupuncture needles increases patients’ pain rating scores and physiological responses [62]. The amygdala controls fear reactions [63], and acupuncture may alter the amygdala-specific brain network [62]. Moreover, some abnormal structures were observed in BD patients’ amygdalae [64]. These findings may support the relatively low proportion of patients with BD who receive acupuncture treatment. Second, Chinese medicine physicians have expertise in treatment, such as acupuncture, TCM prescription, or traumatology. Some patients would say they did not want acupuncture when they came to TCM clinics. Some patients were even unwilling to take TCM. However, we cannot know the actual state of outpatients’ demand in the database study. A more suitable method is to design a questionnaire to ask patients what kind of assistance they want from TCM clinics.

It is essential to detect acupuncture’s positive effect and support its effectiveness on the comorbidities of patients with BD. However, we did not design corresponding research methods to find evidence of the impact of acupuncture on relative symptoms. The current study only showed the characteristics of BD patients who received TCM treatment, principal diagnosis codes, and prescriptions from TCM physicians. Future studies can use results from the current study to research other topics. For example, “myalgia and myositis” was one of the common diagnoses in patients with BD who accepted acupuncture. We can extract patients with BD and “myalgia and myositis” who did and did not use TCM and then compare their usage of corticosteroids and biomarkers of inflammation which were not present in Taiwan’s NHIRD.

### 4.3. Principal Diagnostic Codes Used by TCM Practitioners

In our study, pain-related diseases were the most frequent indication for patients with BD receiving acupuncture. Indeed, pain is one of the most common comorbidities in patients with BD. There exist high rates of pain (29%) and chronic pain (23%) among patients with BD, posing a 2.14 times higher risk of pain in comparison to healthy controls [65]. Patients with BD generally present with various co-occurring pain-related conditions, including arthritis-related pain, rheumatic arthritis, polymyalgia rheumatica, myalgia and myositis, joint pain, back pain, lumbago, migraine, headache, and psychogenic and neuropathic pain [65,66].

Numerous studies have shown that impaired neuroimmune function might be one of the common pathways for pain and mood symptoms in BD [67]. The dysfunction of monoamine neurotransmitters [68], mainly serotonin, noradrenaline, and dopamine, relative metabolites of neurotransmitters as mentioned above [69], and the maladaptive alternations of their receptors in the CNS are involved in the pathogenesis of BD and endogenous pain [70,71]. Therefore, pharmacotherapies acting on monoamine neurotransmitter systems are considered to alleviate pain in affective diseases. Evidence from clinical research supports the ability of tricyclic antidepressants, serotonin–norepinephrine reuptake inhibitors, and certain anticonvulsants to offer pain relief [72]. Clinical studies have proven the therapeutic efficacy of acupuncture for treating 28 conditions, including depression, headache, knee pain, low back pain, neck pain, periarthritis of the shoulder, rheumatoid arthritis, and sprains [73]. Similarly, Chinese herbal medicine can alleviate musculoskeletal pain [74], osteoarthritis [75], fibromyalgia [76], neuropathic pain [77], low back pain [78], and chronic pain [79]. The data from our study demonstrated similar indications to prior studies, suggesting that oral Chinese herbal medicines and acupuncture can treat pain and BD effectively [12,80]. Still, future trials are needed to establish their efficacy further.

Short sleep duration is one of the BD diagnostic criteria, which probably explains why patients with BD are often diagnosed with sleep disturbances. Sleep disturbance may increase the risk of chronic pain [81], which is mediated by cytokines, particularly interleukin (IL)-1β, IL-6, and tumor necrosis factor-alpha (TNF-α) [82]. Elevated IL-6, IL-8, monocyte chemoattractant protein-1, interferon-gamma, and TNF-α levels in patients with BD were associated with inadequate response to antidepressants and short total sleep time [71,83]. Earlier findings indicated that patients with BD have elevated pro-inflammatory cytokine levels and decreased levels of anti-inflammatory cytokines, such as IL-4 and IL-10 [84,85]. The imbalance of pro-inflammatory and anti-inflammatory cytokines in patients with BD may worsen pain perception and sleep quality.

### 4.4. Top Ten Most Commonly Prescribed Chinese Single Herbs and Herbal Formulae

Our study’s most frequently prescribed single herb was Yanhusuo (Corydalis yanhusuo), which was prescribed for myalgia and myositis, lumbago, cough, and headache prescriptions (not shown in the article). Yanhusuo is one of the pain relief and blood activation TCM herbs [86]. The alkaloids from Corydalis binding with GABA, dopamine, and benzodiazepine receptors can alleviate anxiety and depression symptoms [87,88]. A clinical study proved that Yanhusuo significantly decreased pain intensity and bothersomeness scores [89]. Anxiety and depression can increase sensitivity to pain [90], and pain can also lead to anxiety and depression [91], which may explain why Yanhusuo has anti-anxiety, anti-depression, and analgesic effects [86]. Our finding also shows that Shu-Jing-Huo-Xue-Tang was the most frequently prescribed herbal formula in treating lumbago, cough, and headache. It contains 15 herbs, including Dong Gui (Angelica sinensis), Chuan Xiong (Ligusticum wallichii), Di Huang (Rehmannia glutinosa), Fang Feng (Siler divaricatum), Bai Shao (Paeonia lactiflora), Long Dan Cao (Gentiana scabra), Tao Ren (Prunus persica), Huai Niu Xi (Achyranthes bidentata), Fu Ling (Poria cocos), Sheng Jiang (Zingiber officinale), Cang Zhu (Atractylodes lancea), Gan Cao (Glycyrrhiza glabra), Qiang Huo (Notopterygium incisium), Bai Zhi (Angelica anomala), and Chen Pi (Citrus reticulate). In neuropathic rats with chronic constriction injury (a model of neuropathic pain), Shu-Jing-Huo-Xue-Tang achieves anti-hypersensitivity effects by regulating α2 adrenoreceptors [92]. Studies show that activating α2 adrenoreceptors involves pain facilitation by releasing pro-nociceptive transmitters [93]. These findings further support the efficacy of Shu-Jing-Huo-Xue-Tang in alleviating chronic and psychological pain.

Other Chinese herbal medicines with analgesic effects included Danshen (Salvia miltiorrhiza), Shao-Yao-Gan-Cao-Tang, and Xue-Fu-Zhu-Yu-Tang. In our study, Danshen (Salvia miltiorrhiza) was one of the herbs for myalgia and myositis; it has an analgesic effect because of its anti-inflammatory mechanism [94]. Shao-Yao-Gan-Cao-Tang was prescribed for lumbago, headache, and contusion of the ankle and foot excluding toes. Shao-Yao-Gan-Cao-Tang exerts a significant regulatory impact on neuropathic pain, which could increase an individual’s pain threshold and reduce surfactant protein, beta-endorphin, prostaglandin E2, and nitric oxide (NO) levels [95]. In our study, Xue-Fu-Zhu-Yu-Tang was a formula used for headaches; it can relieve distending pain or the tingling sensation accompanying diseases such as ischemic heart disease and arthritis [96].

In our study, the TCM herb and the formula with sedative and hypnotic effects were Suanzaoren (Ziziphus jujube Mill. var. spinosa) and Suan-Zao-Ren-Tang. Suanzaoren is widely used to treat insomnia [97] and possesses anxiolytic effects at low doses and sedative effects at high doses [98]. Danshen (Salvia miltiorrhiza Bge.) and gegen (Pueraria lobata (Willd.) Ohwi) were also prescribed for sleep disturbances in the current study and were associated with a lower risk of depression [99]. The TCM formula Suan-Zao-Ren-Tang contains five herbs, including Suanzaoren (Semen Zizyphi spinosa), Fuling (Sclerotium Poriae Cocos), Chuanxiong (Radix Ligustici Chuanxiong), Zhimu (Rhizoma Anemarrhena), and Gancao (Radix Glycyrrhizae) [100]. Suan-Zao-Ren-Tang ameliorates insomnia through the GABAergic and serotonergic systems and is further suggested to regulate the immune system, particularly inflammation cytokines [101,102]. Other formulae for insomnia include Jia-Wei-Xiao-Yao-San, Chai-Hu-Jia-Long-Gu-Mu-Li-Tang, Gan-Mai-Da-Zao-Tang, and Tian-Wang-Bu-Xin-Dan [103], which seem beneficial for reducing the time required for sleep onset. However, their potential to alleviate insomnia remains uncertain [104].

### 4.5. Strengths and Limitations of the Study

This study is the first study using data from the NHIRD database to explore TCM usage among individuals diagnosed with BD. The major strength of our study is its use of a large population-based claim dataset of one ethnic group, which enabled the analysis of all cases of BD and TCM usage. The results reveal the rationale behind TCM usage in treating patients with BD, providing a new approach among clinicians and a comprehensive understanding of different levels of BD treatments.

To our best acknowledgment, research on the combined use of traditional Chinese medicine (TCM) and conventional medicine (CM) in BD was limited, and the available evidence was not extensive after the literature review. The paper titled “The beneficial effects of the herbal medicine Free and Easy Wanderer Plus (FEWP) for mood disorders: double-blind, placebo-controlled studies” contained in the *Journal of Psychiatric Research*, from 2007, was the only study that mentioned the positive effect of Free and Easy Wanderer Plus add-on with carbamazepine in patients with BD. While there is a rich history of TCM in treating various health conditions, including mental health issues, the integration of TCM with conventional treatments for BD has not been extensively studied in Western scientific literature [19]. Therefore, it is hard to write a review focusing on previous studies showing that TCM added to CM treatment is more effective in treating BD than CM treatment alone.

We are aware that the present study has several limitations. First, we could not determine the BD severity of our study participants without further analysis. The Young Mania Rating Scale can evaluate manic symptoms at baseline and over time in individuals with mania [105]. However, the NHIRD database does not contain such information. We only noticed patients’ current relative ICD-9 code for BD. The possible solution is to compare each BD patient’s outpatient, inpatient, emergency service records, and psychotropic medication usage, which makes it possible to evaluate the prognoses and severities of BD. Second, the current approach could not estimate disease duration without information on the age of onset. Although patients with BD onset before eighteen years old exhibit more severe conditions [51], physicians can try to diagnose BD as early as possible and provide suitable treatments that may relieve BD’s severity. Third, we did not check the prevalence and onset age of BD-related comorbidities or the timing of disease onset. Taking BD comorbid anxiety as an example, a paper published in 2012 indicated that females and early onset age were predominant in patients with bipolar disorder comorbid anxiety [106]. The prior study, which had a small sample size (*n* = 304), might not reveal the prevalence of anxiety in the real world. Still, it told us that women and younger patients had a higher risk of anxiety disorder. Fourth, we could not determine how long patients with BD received TCM treatment after taking standard medications. Therefore, we cannot detect adverse events for the drug–drug interaction between the prescribed medications and internal medicines of TCM directly from the NHIRD but can only observe the relationships between them. For example, there is a need for more studies to consolidate whether the dosages or types of psychotropic medications for BD patients changed or not after TCM intervention under specific controls of NHIRD. Fifth, we cannot confirm the effects of TCM treatments on BD. There are several ways to determine the impact of TCM on BD indirectly. Examples include identifying whether TCM interventions can shorten the BD treatment period, decrease emergency services and acute psychiatric admissions times, or shorten the length of stay days. Sixth, we did not analyze the frequency of using single herbs, herbal formulae, and acupuncture together. The study proposal missed such a key comparison element. TCM clinicians tailor treatments based on their clinical judgment and abilities. We recognize that different TCM practitioners might suggest varied prescriptions for the same conditions, leading to potential differences in symptoms and adding a touch of unpredictability. Finally, we did not tell which combinations of TCM interventions are effective in treating BD.

## 5. Conclusions

In this cohort study, we explored the feasibility of using TCM for patients with BD. Our study revealed the characteristics of patients with BD receiving TCM treatments: a younger average age, a higher percentage of female individuals, more comorbidities of anxiety and alcohol use disorders, and higher mood stabilizer usage rates. Patients with BD visiting TCM clinics were diagnosed with sleep disturbances, joint pain, myalgia, and myositis related to inflammation and neuroimmune diseases by TCM physicians. Yanhusuo and Shu-Jing-Huo-Xue-Tang were the most used prescriptions and, according to a literature review, have suggestive evidence of analgesic effects due to anti-inflammatory mechanisms and modulation of the monoamine neurotransmitter system in the central nervous system in patients with BD.

This study’s findings offer valuable insights for clinical Chinese medicine practitioners, aiding them in selecting appropriate treatment plans when addressing patients with BD. This guidance extends to reflecting on the correlation between the symptoms presented by these patients during medical visits and their chief complaints. Moreover, an analysis of the current status of TCM utilization in BD treatment can serve as a foundational principle for academic and professional education by delving into the specifics of the patterns of single herbs and herbal formulae used by patients with BD since the NNHIRD initiation. This knowledge is a basis for prescribing medications and a foundation for planning additional basic or clinical experiments. Practically, mastering and predicting the development of patients with BD after TCM treatment becomes instrumental in fostering patients’ adherence to medical advice. Furthermore, the adjustment of Western medicine dosage following TCM treatment is anticipated to mitigate toxic side effects.

In conclusion, our study provides new insight into TCM as a potential complementary treatment for BD. However, more evidence-based research is necessary to prove its efficacy in treating BD.

## Figures and Tables

**Figure 1 healthcare-12-00490-f001:**
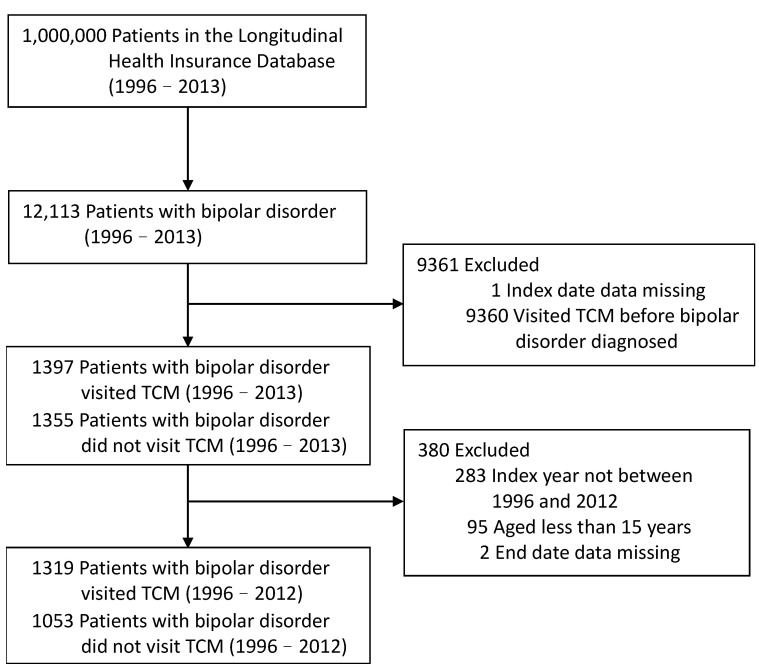
Flowchart of the sample selection procedure.

**Table 1 healthcare-12-00490-t001:** Demographic characteristics and comorbidities of patients with bipolar disorder who did and did not receive traditional Chinese medicine treatment in Taiwan during 1996–2012.

Variable	Total	Non-TCM	TCM	*p*-Value
	N = 2372	*n* = 1053 (44.4)	*n* = 1319 (55.6)	
		Mean ± SD ^a^	Mean ± SD ^a^	
Age (year) *		48.2 ± 19.0	44.3 ± 16.8	<0.0001
	N	*n* (%)	*n* (%)	
15–39	996	410 (38.9)	586 (44.4)	
40–64	942	399 (37.9)	543 (41.2)	
≥65	434	244 (23.2)	190 (14.4)	
Gender *				<0.0001
Female	1091	386 (36.7)	705 (53.4)	
Male	1281	667 (63.3)	614 (46.6)	
Follow-up duration (year) ^a^		8.6 ± 4.8	8.3 ± 4.3	0.1644
Comorbidities *				
Diabetes mellitus	328	149 (14.2)	179 (13.6)	0.6847
Hypertension	576	261 (24.8)	315 (23.9)	0.6097
Hyperlipidemia	376	165 (15.7)	211 (16.0)	0.8282
Ischemic heart disease	74	32 (3.0)	42 (3.2)	0.8397
COPD	492	205 (19.5)	287 (21.8)	0.1716
Anxiety	639	218 (20.7)	421 (31.9)	<0.0001
Attention-deficit disorder	12	5 (0.5)	7 (0.5)	0.8489
Eating disorder, unspecified	3	1(0.1)	2 (0.2)	0.6997
Bulimia nervosa	7	1(0.1)	6 (0.5)	0.1084
Alcoholism	107	35 (3.3)	72 (5.5)	0.0128
Tobacco use	26	7(0.7)	19 (1.4)	0.0714
Obesity	18	6(0.6)	12 (0.9)	0.3431
Medications *				
Lithium carbonate	486	128 (12.2)	358 (27.1)	<0.0001
Carbamazepine	413	141 (13.4)	272 (20.6)	<0.0001
Valproic acid	523	152 (14.4)	371 (28.1)	<0.0001
Lamotrigine	56	21 (2.0)	35 (2.7)	0.2934
Topiramate	26	7 (0.7)	19 (1.4)	0.0714
Aripiprazole	23	0 (0.0)	23 (1.7)	<0.0001
Loxapine	39	21 (2.0)	18 (1.4)	0.2309
Chinese herbal medicine				
Herb	1043	-	1043 (79.1)	-
Formula	1155	-	1155 (87.6)	-
Acupuncture	120	-	120 (9.1)	-

* Chi-square test. ^a^ Student’s *t*-test. Abbreviations: BD, bipolar disorder; COPD, chronic obstructive pulmonary disease; TCM, traditional Chinese medicine.

**Table 2 healthcare-12-00490-t002:** Top 10 indications for traditional Chinese medicine and acupuncture treatments for patients with bipolar disorder in Taiwan during 1996–2012.

No.	Internal Medicine of TCM			Acupuncture		
	ICD-9-CM	Indication	Frequency	ICD-9-CM	Indication	Frequency
1	780.5	Sleep disturbances	470	719.4	Pain in joint	20
2	719.4	Pain in joint	315	729.1	Myalgia and myositis, unspecified	18
3	460	Acute nasopharyngitis	240	724.2	Lumbago	15
4	729.1	Myalgia and myositis, unspecified	236	924.1	Contusion of knee and lower leg	11
5	724.2	Lumbago	212	845.0	Ankle sprain	8
6	786.2	Cough	182	840.9	Sprains and strains of unspecified site of the shoulder and upper arm	7
7	784.0	Headache	180	923.1	Contusion of elbow and forearm	7
8	924.2	Contusion of ankle and foot excluding toe(s)	131	923.2	Contusion of wrist and hand(s) except finger(s) alone	7
9	847.2	Sprain of lumbar	127	844.9	Sprains and strains of unspecified site of the knee and leg	6
10	564.0	Constipation	118	724.5	Backache, unspecified	5

**Table 3 healthcare-12-00490-t003:** Top 10 most common Chinese herbal medicines prescribed to patients with bipolar disorder in Taiwan during 1996–2012.

No.	Single Herb		Herbal Formula	
	Pin-Yin Name (Latin Name)	Frequency	Pin-Yin Name	Frequency
1	Yanhusuo (*Corydalis yanhusuo* W. T. Wang)	347	Shu-Jing-Huo-Xue-Tang	357
2	Danshen (*Salvia miltiorrhiza Bunge*)	297	Jia-Wei-Xiao-Yao-San	347
3	Huangqin (*Scutellaria baicalensis Georgi*)	241	Shao-Yao-Gan-Cao-Tang	312
4	Gegen (*Pueraria lobate*)	215	Suan-Zao-Ren-Tang	261
5	Jiegeng (*Platycodon grandiflorum*)	209	Ge-Gen-Tang	255
6	Gancao (*Glycyrrhiza uralensis Fisch.*)	197	Xue-Fu-Zhu-Yu-Tang	231
7	Suanzaoren (*Ziziphus jujube Mill. var. spinosa)*	197	Tian-Wang-Bu-Xin-Dan	216
8	Duzhong (*Eucommis ulmoidea Oliv.)*	191	Chai-Hu-Jia-Long-Gu-Mu-Li-Tang	213
9	Dahuang (*Rheum palmatum* L.)	189	Gan-Mai-Da-Zao-Tang	210
10	Baizhi (*Angelica dahurica*)	187	Chuan-Xiong-Cha-Tiao-San	207

**Table 4 healthcare-12-00490-t004:** Top 10 most common Chinese herbal medicines prescribed to patients with sleep disturbances after the diagnosis of bipolar disorder in Taiwan during 1996–2012.

No.	Single Herb		Herbal Formula	
	Pin-Yin Name (Latin Name)	Frequency	Pin-Yin Name	Frequency
1	Yejiaoteng (Polygonum multiflorumThunb)	376	Suan-Zao-Ren-Tang	495
2	Yuanzhi (Polygala tenuifolia Willd)	269	Chai-Hu-Jia-Long-Gu-Mu-Li-Tang	404
3	Suanzaoren (Ziziphus jujuba Mill.)	267	Tian-Wang-Bu-Xin-Dan	387
4	Fushen (PoriaCocos Sclerotium)	258	Gan-Mai-Da-Zao-Tang	351
5	Hehuanpi (Albizia julibrissin sensu Baker.)	216	Jia-Wei-Xiao-Yao-San	319
6	Danshen (Salvia miltiorrhiza Bge.)	195	Wen-Dan-Tang	270
7	Botsujen (Platycladus orientalis (L.) Franco)	157	Gui-Pi-Tang	145
8	Gegen (Pueraria lobata (Willd.) Ohwi)	133	Jia-Wei-Xiao-Yao-San	137
9	Huanglian (Coptis chinensis Franch.)	129	Long-Dan-Xie-Gan-Tang	100
10	Shichangpu (Acorus gramineus Soland.)	125	Liu-Wei-Di-Huang-Wan	98

## Data Availability

The datasets used or analyzed during the current study are available from the corresponding author upon reasonable request.

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
