# Peer review of "Usage Patterns of Traditional Chinese Medicine for Patients with Bipolar Disorder: A Population-Based Study in Taiwan"

_healthcare, 2024, doi:10.3390/healthcare12040490_

Round 1

Reviewer 1 Report (Previous Reviewer 2)

Comments and Suggestions for Authors

Most of the comments and suggestions have been addressed in the current revised version, and the article can be accepted after minor revisions.

1.      Line 61. St. The authors mentioned the St. John’s Wort in the Introduction section. St. John's Wort may induce the cytochrome P450 enzymes in the liver, accelerating the metabolism of certain medications. In the case of bipolar disorder, medications such as mood stabilizers (e.g., lithium, valproate), antipsychotics, and antidepressants are commonly prescribed (Line 156, 175, and 254). St. John's Wort may interact with these medications and reduce their efficacy, potentially leading to a worsening of bipolar symptoms. The potential DDI should be noted.

2.      Line 156. While the section mentions "treatment-as-usual medications," it lacks detailed information about the specific medications and their potential impact on the study outcomes. A more explicit discussion of the potential effects of each medication could enhance the understanding of treatment influences.

Author Response

Thank you for your kind review.

1. Line 61. St. The authors mentioned the St. John's Wort in the Introduction section. St. John's Wort may induce the cytochrome P450 enzymes in the liver, accelerating the metabolism of certain medications. In the case of bipolar disorder, medications such as mood stabilizers (e.g., lithium, valproate), antipsychotics, and antidepressants are commonly prescribed (Line 156, 175, and 254). St. John's Wort may interact with these medications and reduce their efficacy, potentially leading to a worsening of bipolar symptoms. The potential DDI should be noted.

Reply: Drug metabolism includes renal and hepatic pathways. Medicines for bipolar disorder, such as lithium, valproate acid, carbamazepine, and antipsychotics, need to monitor serum concentrations strictly[1],[2]. Lithium is renal metabolism, while valproate acid, carbamazepine, and antipsychotics are hepatic metabolism[3],[4],[5]. St. John's Wort can induce cytochrome P450 enzyme activity, accelerating the metabolism of antipsychotics and reducing antipsychotics' control of mania and hypomania[6]. However, there is a lack of effects on valproate acid and carbamazepine of St. John's Wort[7],[8].

Please view the revision on line 58-62 of the revised manuscript.

[1] Dasgupta, A., et al. (2006). "St. John's wort does not interfere with therapeutic drug monitoring of 12 commonly monitored drugs using immunoassays." J Clin Lab Anal 20(2): 62-67.

[2] Grundmann, M., et al. (2014). "Therapeutic drug monitoring of atypical antipsychotic drugs." Acta Pharm 64(4): 387-401.

[3] Ketter, T. A., et al. (1999). "Metabolism and excretion of mood stabilizers and new anticonvulsants." Cell Mol Neurobiol 19(4): 511-532.

[4] Perucca, E. (2002). "Pharmacological and therapeutic properties of valproate: a summary after 35 years of clinical experience." CNS Drugs 16(10): 695-714.

[5] Prior, T. I. and G. B. Baker (2003). "Interactions between the cytochrome P450 system and the second-generation antipsychotics." J Psychiatry Neurosci 28(2): 99-112.

[6] Le, T. T., et al. (2022). "Herb-drug Interactions in Neuropsychiatric Pharmacotherapy - A Review of Clinically Relevant Findings." Curr Neuropharmacol 20(9): 1736-1751.

[7] Le, T. T., et al. (2022). "Herb-drug Interactions in Neuropsychiatric Pharmacotherapy - A Review of Clinically Relevant Findings." Curr Neuropharmacol 20(9): 1736-1751.

[8] Dasgupta, A., et al. (2006). "St. John's wort does not interfere with therapeutic drug monitoring of 12 commonly monitored drugs using immunoassays." J Clin Lab Anal 20(2): 62-67.

2.Line 156. While the section mentions "treatment-as-usual medications," it lacks detailed information about the specific medications and their potential impact on the study outcomes. A more explicit discussion of the potential effects of each medication could enhance the understanding of treatment influences.

Reply: Thank you for your kind review. Conventional psychotropic medications such as mood stabilizers and antipsychotics typically manage bipolar disorder. We selected BD medicines mentioned in Psych Notes- Clinical Pocket Guide 4th Edition by Darlene D. Pedersen [1] and "Taiwan Consensus of Pharmacological Treatment for Bipolar Disorder" by Bai, Y. M., et al. (2013) as treatment-as-usual medications, such as lithium carbonate, carbamazepine, Valproic acid, lamotrigine, topiramate, aripiprazole, or loxapine for further analysis[2]. The medicines mentioned previously are paid for BD medicine treatment by Taiwan's National Health Insurance and are suitable for conducting NHIRD research. All patients with BD received" treatment-as-usual medications" during the study period. The current study was retrospective research, so patients' medical records do not include interference outside NHI during the study period.

Please view the revision on line 168-177 of the revised manuscript.

[1] Pedersen, D. D. PsychNotes: Clinical Pocket Guide, 4th ed.; Publisher: Philadelphia, USA, 2014; pp. 88.

[2] Bai, Y. M., et al. (2013). "Taiwan consensus of pharmacological treatment for bipolar disorder." J Chin Med Assoc 76(10): 547-556.

Reviewer 2 Report (Previous Reviewer 3)

Comments and Suggestions for Authors

The authors have addressed some issues raised during the first round of reviews. I have no further suggestions at this time. 

Author Response

Thank you for your kind review.

Reviewer 3 Report (Previous Reviewer 5)

Comments and Suggestions for Authors

Thank you for inviting me as a reviewer of this valuable manuscript. I recommend following suggestions for improving quality of manuscript.

Introduction Section

(Comment 1) The difference between CAM and TCM is not clear. Please rewrite according to the CAM definition set by NCCIH.

* NCCIH (National Center for Complementary and Integrative Health) classified complementary medicine as follows (https://www.nccih.nih.gov/health/complementary-alternative-or-integrative-health-whats-in-a-name);

(1) Nutritional (e.g., special diets, dietary supplements, herbs, probiotics, and microbial-based therapies).

(2) Psychological (e.g., meditation, hypnosis, music therapies, relaxation therapies).

(3) Physical (e.g., acupuncture, massage, spinal manipulation).

(4) Combinations such as psychological and physical (e.g., yoga, tai chi, dance therapies, some forms of art therapy) or psychological and nutritional (e.g., mindful eating).

Method Section

(Comment 2) What TCM interventions are covered by insurance in the Health Insurance Research Database? (line 86)

(Comment 3) It is important whether the 1,319 people received TCM alone or CM(conventional medicine)+TCM. Are the 1053 people (bipolar disorder patients did not visit TCM) a group that only received CM? (line 129)

Can the results of this study be viewed as a study comparing the ‘CM+TCM group vs CM group’?

Results Section

(Comment 4) The results of this study need to suggest which combinations of TCM interventions are effective in treating bipolar disorder. In clinical practice, we do not provide only a single herb or herbal formula, but also perform a combination of TCM treatments.

(Comment 5) In the case of a study comparing the CM+TCM group vs. the CM group, the TCM treatment effect can be indirectly confirmed by determining whether the bipolar disorder treatment period is shortened. A drawback of this study is that it cannot confirm the effectiveness of TCM in treating bipolar disorder. It is believed that it simply provides trends.

Discussion Section

(Comment 6) There is a need to write a review focusing on previous studies showing that CM+TCM treatment is more effective in treating bipolar disorder than CM treatment alone.

- Suggesting specifically which treatment combinations are effective will help establish future clinical research protocols.

Author Response

Thank you for your kind review.

Introduction Section

(Comment 1) The difference between CAM and TCM is not clear. Please rewrite according to the CAM definition set by NCCIH.

* NCCIH (National Center for Complementary and Integrative Health) classified complementary medicine as follows (https://www.nccih.nih.gov/health/complementary-alternative-or-integrative-health-whats-in-a-name);

(1) Nutritional (e.g., special diets, dietary supplements, herbs, probiotics, and microbial-based therapies).

(2) Psychological (e.g., meditation, hypnosis, music therapies, relaxation therapies).

(3) Physical (e.g., acupuncture, massage, spinal manipulation).

(4) Combinations such as psychological and physical (e.g., yoga, tai chi, dance therapies, some forms of art therapy) or psychological and nutritional (e.g., mindful eating).

Reply: I am more transparent about the definition of CAM through the website—for example, "complementary" means combining the non-mainstream approach and conventional medicine; "alternative" means using the non-mainstream approach instead of conventional medicine. In our study, TCM was the complementary therapy rather than alternative therapy. Although CAM is a broader term encompassing diverse healing practices from various cultural and medical traditions worldwide, CAM treatments can be classified by their primary therapeutic input. How the therapy is taken in or delivered, as follows: nutritional (e.g., special diets, dietary supplements, herbs, and probiotics), psychological (e.g., mindfulness), physical (e.g., massage, spinal manipulation), and combined therapy, such as mental and physical (e.g., yoga, tai chi, acupuncture, dance or art therapies) or psychological and nutritional (e.g., mindful eating).

Please view the revision on line 65-71 of the revised manuscript.

Method Section

(Comment 2) What TCM interventions are covered by insurance in the Health Insurance Research Database? (line 86)

Reply: The National Health Insurance Research Database (NHIRD) covers expenses related to TCM interventions, encompassing oral Chinese herbal medicines (single herbs and herbal formulas), acupuncture, and Chinese traumatology medicine[1].

Please view the revision on line 106-110 of the revised manuscript

[1] Chen, F. P., et al. (2007). "Use frequency of traditional Chinese medicine in Taiwan." BMC Health Serv Res 7: 26.

(Comment 3) It is important whether the 1,319 people received TCM alone or CM(conventional medicine)+TCM. Are the 1053 people (bipolar disorder patients did not visit TCM) a group that only received CM? (line 129)

Can the results of this study be viewed as a study comparing the 'CM+TCM group vs CM group'?

Reply: The 1,319 people in the TCM group received TCM and CM(conventional medicine), and the other 1053 people (bipolar disorder patients who did not visit TCM) in the non-TCM group only received CM.

According to medical visits in chronological order, the 1,319 patients in the TCM group received TCM during CM treatment. The interval between TCM and CM was not the same for each patient. Some were taking TCM and CM simultaneously, some were taking TCM just after CM, and some were taking TCM long after CM. The study roughly compared the "CM plus TCM" and "CM" groups. In further research, we need to control the interval between TCM and CM for a more precise comparison.

Results Section

(Comment 4) The results of this study need to suggest which combinations of TCM interventions are effective in treating bipolar disorder. In clinical practice, we do not provide only a single herb or herbal formula, but also perform a combination of TCM treatments.

Reply: We must state this point in the limitations section, which is absent in the current study. Before presenting the positive effects of TCM combinations on BD, we need to confirm that add-on TCM treatments are better than conventional medicine (CM) for BD. Once we confirm that the TCM add-on CM is much better than CM monotherapy, we can use "top n groups" and "network analysis" to figure out the core herb networks[1], [2]. We have data on each primary indication's top ten TCM herbal and formulae, which may match the top n groups. We also discovered that certain herbals or formulae were prescribed in different primary indications, which can be applied to network analysis to describe the correlation between herbs or formulae and primary indications. Unfortunately, I am not familiar with such analysis methods.

Please view the revision on line 461-462 of the revised manuscript.

[1] Bi, S., et al. (2021). "Detection of Herbal Combinations and Pharmacological Mechanisms of Clinical Prescriptions for Coronary Heart Disease Using Data Mining and Network Pharmacology." Evid Based Complement Alternat Med 2021: 9234984.

[2] Li, Y., et al. (2015). "Herb Network Analysis for a Famous TCM Doctor's Prescriptions on Treatment of Rheumatoid Arthritis." Evid Based Complement Alternat Med 2015: 451319.

(Comment 5) In the case of a study comparing the CM+TCM group vs. the CM group, the TCM treatment effect can be indirectly confirmed by determining whether the bipolar disorder treatment period is shortened. A drawback of this study is that it cannot confirm the effectiveness of TCM in treating bipolar disorder. It is believed that it simply provides trends.

Reply: The point needs to be described in the limitations section, which is the weakness of the current study. There are several ways to determine the effects of TCM in BD indirectly. For example, identifying whether TCM interventions can shorten the BD treatment period, decrease emergency services and acute psychiatric admissions times, or shorten the length of stay days. The preliminary results were that TCM interventions could decrease emergency service times in patients with BD and significantly in those without acute psychiatric admissions. However, TCM interventions did not increase or decrease acute psychiatric hospitalization times or prolong or shorten the length of stay compared to the non-TCM group.[1]

Please view the revision on line 452-456 of the revised manuscript.

[1] Shu Ping Chen, et. al. Psychiatric Services in Patients with Bipolar Disorder Before and After Traditional Chinese Medicine Intervention: A population-based study in Taiwan. The 12th Mind-Body Interface International Stmposium, Taichung, Taiwan, 29th-30th, October, 2022. https://www.mbisymposium.org/2022/upload/eposter/pdf-20221031034433.pdf

Discussion Section

(Comment 6) There is a need to write a review focusing on previous studies showing that CM+TCM treatment is more effective in treating bipolar disorder than CM treatment alone.

- Suggesting specifically which treatment combinations are effective will help establish future clinical research protocols. 

Reply: To my best acknowledgment and after the literature review, research on the combined use of traditional Chinese medicine (TCM) and conventional medicine (CM) in bipolar disorder was limited, and the available evidence was not extensive. The paper titled "The beneficial effects of the herbal medicine Free and Easy Wanderer Plus (FEWP) for mood disorders: double-blind, placebo-controlled studies" contained in the Journal of Psychiatric Research 2007 was the only research that mentioned the positive effect of Free and Easy Wanderer Plus add-on carbamazepine in patients with bipolar disorder[1].

Bipolar disorder is typically managed using conventional psychiatric medications such as mood stabilizers, antipsychotics, and antidepressants. While there is a rich history of TCM in treating various health conditions, including mental health issues, the integration of TCM with conventional treatments for bipolar disorder has not been extensively studied in Western scientific literature. Therefore, it is hard to write a review focusing on previous studies showing that TCM add-on CM treatment is more effective in treating BD than CM treatment alone.

Please view the revision on line 418-429 of the revised manuscript.

[1] Zhang, Z. J., et al. (2007). "The beneficial effects of the herbal medicine Free and Easy Wanderer Plus (FEWP) for mood disorders: double-blind, placebo-controlled studies." J Psychiatr Res 41(10): 828-836.

Reviewer 4 Report (New Reviewer)

Comments and Suggestions for Authors

I think this manuscript was revised enough. I think this manuscript would be suitable for publication in this journal.

Author Response

Thank you for your kind review.

This manuscript is a resubmission of an earlier submission. The following is a list of the peer review reports and author responses from that submission.

Round 1

Reviewer 1 Report

Comments and Suggestions for Authors

The article regards an interesting topic, but contains some concerns which discourage publication.

The study design is inadequate in proving any beneficial use of TCM in the context of BD. With these data, we can only state that a certain percentage of BD patients in Taiwan take TCM, not knowing their precise illness (no worldwide accepted system (DSM-5 or ICD-10) has been employed to establish diagnosis), nor their hospitalization rates, their previous/current medications, their years of disease. A cohort study cannot explore the feasibility of using TCM in patients with BD without analyzing more thorough variables such as described above. Methods are therefore incomplete.
The study is mostly descriptive, it cannot provide any clue or suggestion that TCM is effective in treating BD, either in add on to traditional medications or alone. A large-group observational study should serve this issue, and this is not the case. The fact that patients employed TCM does not mean that their BD symptoms have improved thanks to TCM, further studies are needed which include stratification of TCM/non TCM users in terms of years of illness, disease severity, hospitalization rate, etc. Without these information, no assumption can be made, therefore it serves no purpose but stating that a certain percentage of BD patients in Taiwan takes TCM, which is the only established conclusion that the article can provide.

Author Response

The study design is inadequate in proving any beneficial use of TCM in the context of BD. With these data, we can only state that a certain percentage of BD patients in Taiwan take TCM, not knowing their precise illness (no worldwide accepted system (DSM-5 or ICD-10) has been employed to establish diagnosis), nor their hospitalization rates, their previous/current medications, their years of disease. A cohort study cannot explore the feasibility of using TCM in patients with BD without analyzing more thorough variables such as described above. Methods are therefore incomplete.

Scarce papers discussed the benefit of traditional Chinese medicine (TCM) treatment in bipolar disorder (BD). When we searched PubMed and used keywords ((((bipolar disorder[Title]) OR (mood disorder[Title])) OR (mood disorders[Title]))) AND ((((acupuncture[Title]) OR (herbal[Title])) OR (complementary[Title]) OR (traditional Chinese medicine[Title])))) ) before April 6th, 2022, we only found 21 papers. After clicking the "clinical trial" box, only one paper met the criteria, which title was "The beneficial effects of the herbal medicine Free and Easy Wanderer Plus (FEWP) for mood disorders: double-blind, placebo-controlled studies" contained in the Journal of Psychiatric Research 2008. Perhaps the reason is that few patients with BD use TCM. Identifying the BD patients' population using TCM was the primary task. As TCM is an important part of complementary and alternative medicine (CAM), we carefully read 20 other papers to prove this suspect (few patients with BD use TCM). We acknowledge that four papers analyzed BD patients' intention to receive CAM. Summaries were as follows. Two papers identified the population of BD patients using CAM. Fifty percent of BD patients took vitamins or herbs[1], and 44% of elderly BD patients used CAM[2]. Worthy of attention, most BD patients didn't discuss their CAM usage with their physicians[3]. BD patients receiving CAM thought CAM treatment was cheaper than mental health services[4]. BD patients were more likely to perceive that psychotropic medications were ineffective at relieving manic or depressive symptoms. Still, medication compliance was not significantly associated with using complementary alternative medicine[5]. The four previous papers didn't describe which herbals BD patients had taken. We then read relevant review papers to filter identity herbals. For example, St. John's wort (Hypericum perforatum) can relieve the prognoses of depression but increase the risk of mania[6]. A reviewed paper published in 2014, "Clinical applications of herbal medicines for anxiety and insomnia; targeting patients with bipolar disorder," indicated few studies conducted on herbal usage in BD. The authors collected the literature on the herbal treatment of insomnia and anxiety for BD patients more likely to suffer anxiety and insomnia between mood episodes. They presumed the potential clinical applications of herbal treatment for these symptoms in patients with bipolar disorder[7]. Various literature evidence indicated a population-based study in BD patients using TCM is necessary. Depending on the reasons mentioned above, we decided to investigate the BD population using TCM, what diseases BD patients go to see TCM physicians, and what kinds of TCM BD patients were prescribed using Taiwan's national health insurance database (NHIRD).

[1] Kilbourne AM, Copeland LA, Zeber JE, Bauer MS, Lasky E, Good CB. Determinants of complementary and alternative medicine use by patients with bipolar disorder. Psychopharmacol Bull. 2007;40(3):104-15.

[2] Keaton D, Lamkin N, Cassidy KA, Meyer WJ, Ignacio RV, Aulakh L, Blow FC, Sajatovic M. Utilization of herbal and nutritional compounds among older adults with bipolar disorder and with major depression. Int J Geriatr Psychiatry. 2009 Oct;24(10):1087-93.

[3] Keaton D, Lamkin N, Cassidy KA, Meyer WJ, Ignacio RV, Aulakh L, Blow FC, Sajatovic M. Utilization of herbal and nutritional compounds among older adults with bipolar disorder and with major depression. Int J Geriatr Psychiatry. 2009 Oct;24(10):1087-93.

[4] Perron BE, Jarman CN, Kilbourne AM. Access to conventional mental health and medical care among users of complementary and alternative medicine with bipolar disorder. J Nerv Ment Dis. 2009 Apr;197(4):287-90.

[5] Jarman CN, Perron BE, Kilbourne AM, Teh CF. Perceived treatment effectiveness, medication compliance, and complementary and alternative medicine use among veterans with bipolar disorder. J Altern Complement Med. 2010 Mar;16(3):251-5.

[6] Andreescu C, Mulsant BH, Emanuel JE. Complementary and alternative medicine in the treatment of bipolar disorder--a review of the evidence. J Affect Disord. 2008 Sep;110(1-2):16-26.

[7] Baek JH, Nierenberg AA, Kinrys G. Clinical applications of herbal medicines for anxiety and insomnia; targeting patients with bipolar disorder. Aust N Z J Psychiatry. 2014 Aug;48(8):705-15.

The study is mostly descriptive, it cannot provide any clue or suggestion that TCM is effective in treating BD, either in add on to traditional medications or alone. A large-group observational study should serve this issue, and this is not the case. The fact that patients employed TCM does not mean that their BD symptoms have improved thanks to TCM, further studies are needed which include stratification of TCM/non TCM users in terms of years of illness, disease severity, hospitalization rate, etc. Without these information, no assumption can be made, therefore it serves no purpose but stating that a certain percentage of BD patients in Taiwan takes TCM, which is the only established conclusion that the article can provide.

There are several limits of NHIRD. This database doesn't contain the Young Mania Rating Scale to evaluate manic symptoms at baseline and over time in individuals with mania. We only noticed patients' current relative ICD-9 code for BD. We can't identify the exact BD onset time but refer to the new BD occurrence time. Depending on the changes in BD patients' outpatient, inpatient, emergency service records, and psychotropic medication usage, it is possible to assess the prognoses and severities of BD. When TCM treatments intervene in BD prognoses, we can compare the differences in psychiatric services before and after TCM to analyze the effectiveness of TCM. The previous thoughts needed different study methods to prove, and we had preliminary results but revealed them in the other topic.

Reviewer 2 Report

Comments and Suggestions for Authors

1.      Line 43-44. 1. Introduction “Several papers indicated…” The authors should add the cited references.

2.      Line 74 2.2 Study population. The exclusion criteria are index year, age range, and end date data. Did the authors consider the inclusion criteria, such as a. gender: the male/female ratio, b. the age upper bound of the patients, c. polypharmacy, especially the medications other than “treatment-as-usual medications”, c. consciousness or not, d. history of diseases, etc.

3.      Line 113. 3.1 Demographic characteristics. Table1. Why did the 12 diseases are included as comorbidities?

4.      Line 121. 3.2 TCM treatment interventions. Why do the authors mention Chinese traumatology here?

5.      Line 141. 3.3 Principal diagnostic code used by TCM practitioners. Did the authors find any AE for the DDI between the prescribed medications and the internal medicine of TCM?

6.      Line 156. 3.4 Top ten most commonly prescribed Chinese single herbs and herbal formulae. Did the authors find any differences between the use of single herbs and herbal formulae?

7.      Line 176. 4. Discussion. The authors concluded that “the patients with BD may relate to the severity of anxiety”. Did the authors check the onset age of BD and anxiety, and determine the relationship between the two?

Author Response

1. Line 43-44. 1. Introduction “Several papers indicated…” The authors should add the cited references.

Traditional Chinese Medicine (TCM) is integral to CAM and treatment options for BD patients.

2.      Line 74 2.2 Study population. The exclusion criteria are index year, age range, and end date data. Did the authors consider the inclusion criteria, such as a. gender: the male/female ratio, b. the age upper bound of the patients, c. polypharmacy, especially the medications other than “treatment-as-usual medications”, c. consciousness or not, d. history of diseases, etc.

Age at onset (AAO) is a critical variable in the prognoses of BD[1]. There were several cut-off BD onset ages in the literature. For example, forty years old is a cut-off for early-onset and late-onset bipolar disorder (EOBD and LOBD)[2]. Approximately 30-60% of BD patients onset between ages 15-19[3], called pediatric or juvenile BD[4]. Patients who suffered or maintained BD over 50 years old named older age bipolar disorder (OABD)[5]. Bolton et al. 2021 systematically searched the databases. They proposed the trimodal age-at-onset distribution by average early-onset age was seventeen, mid-onset age was twenty-six, and late-onset age was forty-six[6]. Depending on the previous literature statements and citizens over 65 years old in Taiwan being elderly, we decided to include BD patients more than fifteen years old.

3.      Line 113. 3.1 Demographic characteristics. Table1. Why did the 12 diseases are included as comorbidities?

Physical diseases are risk factors for psychiatric disorders[7],[8],[9]; otherwise, psychiatric disorders also affect physical illnesses. When patients with physical conditions or psychiatric disorders do not control well, both will increase the disease prognoses bilaterally[10]. Over ninety percent of patients with BD experience recurrence during their lifetimes[11], and the one-year recurrence rate ranges from 35% to 57%[12]. Cognitive functions, pharmacotherapies, and psychiatric and medical comorbidities of patients with BD affect recurrence rates, increasing the odds of suicidality, disability, unemployment, and re-hospitalization[13]. According to previous statements, we selected the following comorbidities in BD patients based on the literature review. Physical comorbidities in BD patients included hypertension[14], diabetes mellitus[15], hyperlipidemia[16], obesity[17], tobacco use[18], and chronic obstructive pulmonary disease (COPD)[19],[20],[21]. The previous comorbid diseases were risk factors for cardiovascular disease, such as ischemic heart disease (IHD)[22], and one of the most common premature mortalities among BD patients.[23],[24] Patients with BD also comorbid one to several psychiatric diseases. A meta-analysis study indicated that 40% of patients with BD suffer from any anxiety disorder[25]. Seventeen percent of BD patients had attention-deficit/hyperactivity disorder (ADHD)[26]. McElroy et al. surveyed 875 patients about their eating disorder comorbidities, and 14.3% had at least one comorbid lifetime eating disorder, such as bulimia nervosa[27]. Regier et al. found 43.6% of patients with BD had an alcohol diagnosis[28].

4.      Line 121. 3.2 TCM treatment interventions. Why do the authors mention Chinese traumatology here?

NHIRD pays for TCM treatments such as oral Chinese herbal medicines (single herbs and herbal formulas), acupuncture, and Chinese traumatology medicine. We only analyzed Chinese herbal medicine and acupuncture. It is better to explain the materials and methods but not the results part.

5.      Line 141. 3.3 Principal diagnostic code used by TCM practitioners. Did the authors find any AE for the DDI between the prescribed medications and the internal medicine of TCM?

We cannot detect adverse events for the drug-drug interaction between the prescribed medications and internal medicines of TCM directly from the NHIRD but only observe the relationships between each other. For example, whether the dosages or types of psychotropic medications for BD patients changed or not after TCM intervention under specific controls of NHIRD needs more studies to consolidate.

6.      Line 156. 3.4 Top ten most commonly prescribed Chinese single herbs and herbal formulae. Did the authors find any differences between the use of single herbs and herbal formulae?

TCM physicians prescribe Chinese herbal medicine with one or more herbal formulas combined with several herbs based on the patient's symptoms. One herbal formula contains several single herbs.

7.      Line 176. 4. Discussion. The authors concluded that “the patients with BD may relate to the severity of anxiety”. Did the authors check the onset age of BD and anxiety, and determine the relationship between the two?

We didn't check the onset of anxiety. In other words, we couldn't indicate whether BD patients included in the study had anxiety disorders before or after the BD diagnoses.

[1] Bolton, S., et al. (2021). "Bipolar disorder: Trimodal age-at-onset distribution." Bipolar Disord 23(4): 341-356.

[2] Lavin, P., et al. (2022). "Clinical correlates of late-onset versus early-onset bipolar disorder in a global sample of older adults." Int J Geriatr Psychiatry 37(12).

[3] Goetz, M., et al. (2015). "Early stages of pediatric bipolar disorder: retrospective analysis of a Czech inpatient sample." Neuropsychiatr Dis Treat 11: 2855-2864.

[4] Yee, C. S., et al. (2019). "Maintenance Pharmacological Treatment of Juvenile Bipolar Disorder: Review and Meta-Analyses." Int J Neuropsychopharmacol 22(8): 531-540.

[5] Szmulewicz, A., et al. (2020). "Longitudinal analysis of cognitive performances in recent-onset and late-life Bipolar Disorder: A systematic review and meta-analysis." Bipolar Disord 22(1): 28-37.

[6] Bolton, S., et al. (2021). "Bipolar disorder: Trimodal age-at-onset distribution." Bipolar Disord 23(4): 341-356.

[7] Chien IC, Lin CH, Chou YJ, Chou P. Risk of hypertension in patients with bipolar disorder in Taiwan: a population-based study. Comprehensive psychiatry 2013;54(6):687-93.

[8] Hsu JH, Chien IC, Lin CH. Increased risk of hyperlipidemia in patients with bipolar disorder: a population-based study. Gen Hosp Psychiatry 2015;37(4):294-8.

[9] Su VY, Hu LY, Yeh CM, Chiang HL, Shen CC, Chou KT, et al. Chronic obstructive pulmonary disease associated with increased risk of bipolar disorder. Chronic respiratory disease 2017;14(2):151-60.

[10] Penninx B, Lange SMM. Metabolic syndrome in psychiatric patients: overview, mechanisms, and implications. Dialogues in clinical neuroscience 2018;20(1):63-73.

[11] Solomon DA, Keitner GI, Miller IW, Shea MT, Keller MB. Course of illness and maintenance treatments for patients with bipolar disorder. J Clin Psychiatry 1995;56:5-13

[12] Shim, I. H., et al. (2017). "Predictors of a Shorter Time to Hospitalization in Patients with Bipolar Disorder: Medication during the Acute and Maintenance Phases and Other Clinical Factors." Clin Psychopharmacol Neurosci 15(3): 248-255.

[13] Peters AT, West AE, Eisner L, Baek J, Deckersbach T. The burden of repeated mood episodes in bipolar I disorder: Results from the national epidemiological survey on alcohol and related conditions. J Nerv Ment Dis 2016;204:87-94.

[14] Goldstein BI, Fagiolini A, Houck P, Kupfer DJ. Cardiovascular disease and hypertension among adults with bipolar I disorder in the United States. Bipolar disorders 2009;11(6):657-62.

[15] Cassidy F, Ahearn E, Carroll BJ. Elevated frequency of diabetes mellitus in hospitalized manic-depressive patients. The American journal of psychiatry 1999;156(9):1417-20.

[16] Kilbourne A, Cornelius J, Han X, Pincus H, Shad M, Salloum I, et al. Burden of General Medical Conditions among Individuals with Bipolar Disorder. Bipolar disorders 2004;6:368-73.

[17] Fagiolini A, Frank E, Houck PR, Mallinger AG, Swartz HA, Buysse DJ, et al. Prevalence of obesity and weight change during treatment in patients with bipolar I disorder. The Journal of clinical psychiatry 2002;63(6):528-33.

[18] Lasser K, Boyd JW, Woolhandler S, Himmelstein DU, McCormick D, Bor DH. Smoking and Mental IllnessA Population-Based Prevalence Study. JAMA 2000;284(20):2606-10.

[19] Chen, W., et al. (2015). "Risk of cardiovascular comorbidity in patients with chronic obstructive pulmonary disease: a systematic review and meta-analysis." Lancet Respir Med 3(8): 631-639.

[20] Hsu, J. H., et al. (2017). "Increased risk of chronic obstructive pulmonary disease in patients with bipolar disorder: A population-based study." J Affect Disord 220: 43-48.

[21] Su, V. Y., et al. (2017). "Chronic obstructive pulmonary disease associated with increased risk of bipolar disorder." Chron Respir Dis 14(2): 151-160.

[22] Hsu, J. H., et al. (2021). "Increased risk of ischemic heart disease in patients with bipolar disorder: A population-based study." J Affect Disord 281: 721-726.

[23] Ösby U, Brandt L, Correia N, Ekbom A, Sparén P. Excess Mortality in Bipolar and Unipolar Disorder in Sweden. Arch Gen Psychiatry 2001;58(9):844-50.

[24] Chan, J. K. N., et al. (2021). "Excess mortality and life-years lost in people with bipolar disorder: an 11-year population-based cohort study." Epidemiol Psychiatr Sci 30: e39.

[25] Yapici Eser H, Kacar AS, Kilciksiz CM, Yalçinay-Inan M, Ongur D. Prevalence and Associated Features of Anxiety Disorder Comorbidity in Bipolar Disorder: A Meta-Analysis and Meta-Regression Study. Frontiers in psychiatry 2018;9(229).

[26] Schiweck C, Arteaga-Henriquez G, Aichholzer M, Edwin Thanarajah S, Vargas-Cáceres S, Matura S, et al. Comorbidity of ADHD and adult bipolar disorder: A systematic review and meta-analysis. Neuroscience and biobehavioral reviews 2021;124:100-23.

[27] McElroy, S. L., et al. (2011). "Prevalence and correlates of eating disorders in 875 patients with bipolar disorder." J Affect Disord 128(3): 191-198.

[28] Regier, D. A., et al. (1990). "Comorbidity of mental disorders with alcohol and other drug abuse. Results from the Epidemiologic Catchment Area (ECA) Study." JAMA 264(19): 2511-2518.

Reviewer 3 Report

Comments and Suggestions for Authors

This is an interesting, highly relevant paper on the use of TCM among patients with bipolar disorder. The methods are well-described and the paper  is, overall, well-written. I do think it could use a review of some aspects of the English language, but I think the issue is overall minor. I did notice the use, at times, of non-inclusive language ("bipolar disorder patients" instead of "patients with bipolar disorder" and I suggest reviewing that. 

The finding of higher rates of use of TCM among younger patients is a little counterintuitive to me, as I would expect the opposite. I understood the authors' explanation about a possible effect of severity, and that does make sense, but I was hoping there were another proxy measure for severity that could be included in the analysis (e.g., number of outpatient visits, number of admissions), to verify that hypothesis. 

Author Response

This is an interesting, highly relevant paper on the use of TCM among patients with bipolar disorder. The methods are well-described and the paper  is, overall, well-written. I do think it could use a review of some aspects of the English language, but I think the issue is overall minor. I did notice the use, at times, of non-inclusive language ("bipolar disorder patients" instead of "patients with bipolar disorder" and I suggest reviewing that.

Thank you. We will change "bipolar disorder patients" to "patients with bipolar disorder." "Patients with bipolar disorder" is a more neutral word than "bipolar disorder patients."

The finding of higher rates of use of TCM among younger patients is a little counterintuitive to me, as I would expect the opposite. I understood the authors' explanation about a possible effect of severity, and that does make sense, but I was hoping there were another proxy measure for severity that could be included in the analysis (e.g., number of outpatient visits, number of admissions), to verify that hypothesis.

At the end of 2013, 6,738,483 (30.90%) valid beneficiaries of the National Health Insurance in Taiwan had used TCM of the 21,808,429 total outpatients during the year. Among TCM users, females were higher than males (female: male = 1.47:1). The age distribution peaked at 25 to 44 (36.51%), followed by 45-64 (29.29%)[1]. Depending on the changes in BD patients' outpatient, inpatient, emergency service records, and psychotropic medication usage, it is possible to assess the prognoses and severities of BD. When TCM treatments intervene in BD prognoses, we can compare the differences in psychiatric services before and after TCM to analyze the effectiveness of TCM. The previous thoughts needed different study methods to prove, and we had preliminary results but revealed them in the other topic.

[1] Statistics of Medical Care, National Health Insurance 2013

Reviewer 4 Report

Comments and Suggestions for Authors

1. In my opinion, this study is not about TCM for BD, but rather an analysis of patients with BD who received TCM.  It would be good to clarify this in the introduction, discussion, and conclusion.

2. It is necessary to explain why the authors excluded from the analysis who visited TCM before BD diagnosis.

3. Isn't it possible that there were many women, young people, and patients with anxiety by chance?

4. It is necessary to analyze the frequency of using single herb, herbal formulae and/or acupuncture together. In addition, it is necessary to analyze the single herb and herbal formula used repeatedly in one person.

5. Since the authors divided the drugs into single herb and herbal formula, it would be better to analyze the indications of internal medicine by dividing them into single herb and herbal formula.

6. Can the low proportion of acupuncture treatment be explained only by patient’s fear of pain? Are there any possibilities according to the doctor's treatment preference?

7. In the ‘4.3. Principal diagnostic codes used by TCM practitioners’ section, evidences on the effect of acupuncture on the symptoms included is needed.

8. Anxiety and depression increase pain sensitivity, but pain can also lead to anxiety and depression. Considering the traditional use of Yanhusuo, wouldn't it be that Yanhusuo affects anxiety and depression through the analgesic effect rather than having an analgesic effect through alleviating anxiety and depression symptoms?

9. Since there are many pain-related symptoms in the TCM indication, it seems necessary to explain the relationship between BD and pain.

Author Response

1. In my opinion, this study is not about TCM for BD, but rather an analysis of patients with BD who received TCM.  It would be good to clarify this in the introduction, discussion, and conclusion.

1. Thank you for your guidance. It is essential to explain that this study aims to analyze patients with BD who received TCM but not TCM for BD treatments.

2. It is necessary to explain why the authors excluded from the analysis who visited TCM before BD diagnosis.

2. Taiwan's longitudinal health insurance database 2000 (LHID2000) contains all the original claim data of one million individuals randomly sampled from the 2000 Registry for Beneficiaries of NHIRD. Seventy-seven percent (n= 9,361) of 12,113 patients with BD of LHID 2000 received TCM. To avoid potential TCM interference before the diagnosis of bipolar disorder, we were more curious to explore the phenomena that patients who never received TCM treatment before the diagnosis of BD changed to TCM treatment but still maintained psychiatric therapies. Therefore we excluded patients with BD who received TCM before their diagnoses of BD.

3. Isn't it possible that there were many women, young people, and patients with anxiety by chance?

In our study, we cannot analyze the prevalence of anxiety. A paper published in 2012 indicated that female and early onset age were predominant in patients with bipolar one disorder comorbid anxiety[1]. The prior literature, which had small subjects, might not reveal the prevalence of anxiety in the real world. Still, it told us that women and younger patients had a higher risk of anxiety disorder.

[1] Tsai, H. C., et al. (2012). "Empirically derived subgroups of bipolar I patients with different comorbidity patterns of anxiety and substance use disorders in Han Chinese population." J Affect Disord 136(1-2): 81-89.

4. It is necessary to analyze the frequency of using single herb, herbal formulae and/or acupuncture together. In addition, it is necessary to analyze the single herb and herbal formula used repeatedly in one person.

We appreciate your attention to this matter. It's worth noting that the study proposal, unfortunately, omitted this crucial comparison element. TCM clinicians offer appropriate treatments according to their clinical reasons and capabilities. It's important to acknowledge that various TCM practitioners may present diverse prescriptions for identical ailments, introducing potential variations in symptoms and contributing to heightened factors of instability.

5. Since the authors divided the drugs into single herb and herbal formula, it would be better to analyze the indications of internal medicine by dividing them into single herb and herbal formula.

It is a very precious advice. We did a more in-depth study of this comment. The results are shown in the attachment table.

6. Can the low proportion of acupuncture treatment be explained only by patient’s fear of pain? Are there any possibilities according to the doctor's treatment preference?

Thank you for the advice. Patients' fear of pain was not the only reason for lower percentages of receiving acupuncture than taking traditional Chinese medicine. Chinese medicine physicians have expertise in treatment, such as acupuncture, TCM prescription, or traumatology. Some patients would say they did not want acupuncture when they came to TCM clinics. Some patients were even unwilling to take TCM. However, we cannot know the actual state of outpatients' demand in the database study. A more suitable method is to design a questionnaire to ask patients what kind of assistance they want from TCM clinics.

7. In the ‘4.3. Principal diagnostic codes used by TCM practitioners’ section, evidences on the effect of acupuncture on the symptoms included is needed.

It is essential to detect acupuncture's positive effect and support its effectiveness. The study aimed to analyze the characteristics of BD patients who received TCM treatment, principal diagnosis codes, and prescriptions from TCM physicians. Therefore, we didn't design corresponding research methods to find evidence of the effect of acupuncture on relative symptoms. We can still use results from the current study to research other topics. For example, "myalgia and myositis" was one of the common diagnoses in patients with BD who accepted acupuncture. We can extract patients with BD and "myalgia and myositis" who did and didn't use TCM, then compare their usage of corticosteroids and biomarkers of inflammation which didn't present in Taiwan's NHIRD.

8. Anxiety and depression increase pain sensitivity, but pain can also lead to anxiety and depression. Considering the traditional use of Yanhusuo, wouldn't it be that Yanhusuo affects anxiety and depression through the analgesic effect rather than having an analgesic effect through alleviating anxiety and depression symptoms?

Anxiety and depression increase pain sensitivity; pain can also aggravate anxiety and depression conditions. According to materia medica books, Yanhusuo's (Corydalis yanhusuo) primary effect includes pain relief and blood activation.

9. Since there are many pain-related symptoms in the TCM indication, it seems necessary to explain the relationship between BD and pain.

Pain is one of the most common comorbidities in bipolar disorder (BD) patients. There exist high rates of pain (29%) and chronic pain (23%) among BD patients, posing a 2.14 times higher risk of pain than healthy controls[1]. BD patients are generally presented with various co-occurring pain-related conditions, including arthritis-related pain, rheumatic arthritis, polymyalgia rheumatica, myalgia and myositis, joint pain, back pain, lumbago, migraine, headache, psychogenic and neuropathic pain[2], [3].

        Numerous studies have shown that impaired neuroimmune function might be one of the common pathways for pain and mood symptoms in BD[4]. The dysfunction of monoamine neurotransmitters[5], mainly serotonin, noradrenaline, and dopamine, their metabolites[6], and the maladaptive alternations of their receptors in the CNS[7]get involved in the pathogenesis of BD and endogenous pain[8]. Therefore, pharmacotherapies acting on monoamine neurotransmitter systems are considered to alleviate pain in affective diseases. Evidence from clinical research supports the ability of tricyclic antidepressants, serotonin-norepinephrine reuptake inhibitors, and certain anticonvulsants to offer pain relief[9].

[1] Stubbs, B.; Eggermont, L.; Mitchell, A.J.; De Hert, M.; Correll, C.U.; Soundy, A.; Rosenbaum, S.; Vancampfort, D. The prevalence of pain in bipolar disorder: a systematic review and large-scale meta-analysis. Acta psychiatrica Scandinavica 2015, 131, 75-88, doi:10.1111/acps.12325.

[2] Birgenheir, D.G.; Ilgen, M.A.; Bohnert, A.S.B.; Abraham, K.M.; Bowersox, N.W.; Austin, K.; Kilbourne, A.M. Pain conditions among veterans with schizophrenia or bipolar disorder. General Hospital Psychiatry 2013, 35, 480-484, doi:https://doi.org/10.1016/j.genhosppsych.2013.03.019.

[3] Leo, R.J.; Singh, J. Migraine headache and bipolar disorder comorbidity: A systematic review of the literature and clinical implications. Scandinavian journal of pain 2016, 11, 136-145, doi:10.1016/j.sjpain.2015.12.002.

[4] Goesling, J.; Lin, L.A.; Clauw, D.J. Psychiatry and Pain Management: at the Intersection of Chronic Pain and Mental Health. Current psychiatry reports 2018, 20, 12, doi:10.1007/s11920-018-0872-4.

[5] Manji, H.K.; Quiroz, J.A.; Payne, J.L.; Singh, J.; Lopes, B.P.; Viegas, J.S.; Zarate, C.A. The underlying neurobiology of bipolar disorder. World psychiatry : official journal of the World Psychiatric Association (WPA) 2003, 2, 136-146.

[6] Kurita, M. Noradrenaline plays a critical role in the switch to a manic episode and treatment of a depressive episode. Neuropsychiatric disease and treatment 2016, 12, 2373-2380, doi:10.2147/ndt.s109835.

[7] Lee, M., C. Schwab, and P.L. McGeer, Astrocytes are GABAergic cells that modulate microglial activity. Glia, 2011. 59(1): 152-65.

[8] Benedetti, F., et al., Higher Baseline Proinflammatory Cytokines Mark Poor Antidepressant Response in Bipolar Disorder. J Clin Psychiatry, 2017. 78(8): p. e986-e993.

[9] Macone, A. and J.A.D. Otis, Neuropathic Pain. Semin Neurol, 2018. 38(6): p. 644-653.

Reviewer 5 Report

Comments and Suggestions for Authors

Thank you for inviting me as a reviewer of this valuable manuscript. I recommend following suggestions for improving the quality of manuscript.

Introduction Section

(Comment 1) Introduction Section is too short to explain background and aim of this study. I recommend authors to add (1) conventional medicine treatment for bipolar disorder, (2) CAM treatment for bipolar disorder, (3) difference between TCM and CAM

Method

(Comment 2) I recommend authors to supplement citation of data source. (line 55)

(Comment 3) I recommend authors to add types of TCM treatments in TCM clinics. (line 64)

- Are there only 3 types of TCM interventions (herb, herbal medicine and acupuncture) in the data? Do TCM clinics only provide 3 treatments? This should be stated in the manuscript (Method or Discussion).

Discussion Section

(Comment 4) I recommend authors to supplement policy development point or suggestion for future research based on the result.

Author Response

Introduction Section

(Comment 1) Introduction Section is too short to explain background and aim of this study. I recommend authors to add (1) conventional medicine treatment for bipolar disorder, (2) CAM treatment for bipolar disorder, (3) difference between TCM and CAM

(1) Bipolar disorder includes recurrent periods of mania, hypomania, and depression. Mood stabilizers are approved to suppress shifts between mania and depression. Monotherapy of mood stabilizers should be the primary choice, at least in mild and moderate mania. Although poly-therapy has proven to be more effective with the combinations of mood stabilizers plus atypical antipsychotics, poly-therapy should be reserved for severe mania or as a subsequent step in mild and moderate mania after unsuccessful monotherapy[1]. Monotherapy for mania and hypomania episodes includes Lithium, valproate, aripiprazole, olanzapine, quetiapine, risperidone, ziprasidone, and carbamazepine. Combinations of mania and hypomania have Lithium plus valproate, Lithium or valproate plus aripiprazole or olanzapine or quetiapine or risperidone or ziprasidone. Monotherapy for depressive episodes includes Quetiapine, lamotrigine, valproate, and Lithium. Combinations of depressive episodes have valproate plus Lithium, Lithium or valproate plus lamotrigine, and Lithium or valproate plus Venlafaxine or L-Thyroxine or Topiramate[2].

(2) Research on complementary and alternative treatments for depression or bipolar depression is more than bipolar disorder. Complementary and alternative therapies for bipolar disorder include omega-3 fatty, St. John's wort (Hypericum perforatum), S-adenosyl-L-methionine (SAMe), and acupuncture. St. John's wort and S-adenosyl-L-methionine can relieve mild to moderate depression but have the potential to induce mania. Evidence regarding the benefits of omega-3 fatty acids or acupuncture is inconsistent[3].

(3) Complementary and Alternative Medicine is a broader term encompassing diverse healing practices from various cultural and medical traditions worldwide, such as traditional Chinese medicine (TCM), Ayurveda, homeopathy, naturopathy, etc. TCM has its roots in ancient Chinese philosophy, including principles such as Yin and Yang and the concept of Qi (vital energy), whose treatments include acupuncture, herbal medicine, massage (tui na), and qigong.

[1]  World J Biol Psychiatry, 2018. 19(1): p. 2-58.

[2] Bai, Y. M., et al. (2013). "Taiwan consensus of pharmacological treatment for bipolar disorder." J Chin Med Assoc 76(10): 547-556.

[3] Andreescu, C., et al. (2008). "Complementary and alternative medicine in the treatment of bipolar disorder--a review of the evidence." J Affect Disord 110(1-2): 16-26.

Method

(Comment 2) I recommend authors to supplement citation of data source. (line 55)

(Comment 3) I recommend authors to add types of TCM treatments in TCM clinics. (line 64)

- Are there only 3 types of TCM interventions (herb, herbal medicine and acupuncture) in the data? Do TCM clinics only provide 3 treatments? This should be stated in the manuscript (Method or Discussion).

Comment 2

The structure of the NHIRD files is described in detail on the NHIRD website and in other publications[1].

Comment 3

Each traditional Chinese medicine (TCM) practitioner specializes in various areas, including qigong, dietary therapy, etc. However, the National Health Insurance Research Database (NHIRD) covers expenses related to TCM interventions, encompassing oral Chinese herbal medicines (single herbs and herbal formulas), acupuncture, and Chinese traumatology medicine[2].

[1] Chang, L. C., et al. (2008). "Utilization patterns of Chinese medicine and Western medicine under the National Health Insurance Program in Taiwan, a population-based study from 1997 to 2003." BMC Health Serv Res 8: 170.

[2] Chang, L. C., et al. (2008). "Utilization patterns of Chinese medicine and Western medicine under the National Health Insurance Program in Taiwan, a population-based study from 1997 to 2003." BMC Health Serv Res 8: 170.

Discussion Section

(Comment 4) I recommend authors to supplement policy development point or suggestion for future research based on the result.

    The study's findings offer valuable insights for clinical Chinese medicine practitioners, aiding them in selecting appropriate treatment plans when addressing patients with bipolar disorders. This guidance extends to reflecting on the correlation between the symptoms presented by these patients during medical visits and their chief complaints. Moreover, an analysis of the current status of traditional Chinese medicine utilization in bipolar disorder treatment can serve as a foundational principle for both academic and professional education.

    By delving into the specifics, understanding the patterns of single herbs and herbal formulas used by patients with bipolar disorder since the initiation of the National Health Insurance becomes crucial. This knowledge serves as a basis for prescribing medications and as a foundation for planning additional basic or clinical experiments.

    In practical terms, mastering and predicting the development of patients with bipolar disorder post-traditional Chinese medicine treatment becomes instrumental in fostering patients' adherence to medical advice. Furthermore, the adjustment of Western medicine dosage following traditional Chinese medicine treatment is anticipated to mitigate toxic side effects.

    Considering a holistic approach, the combined utilization of traditional Chinese and Western medicine in treating bipolar disorders demonstrates potential. This approach can place patients in a relatively stable condition, simultaneously reducing societal healthcare costs.